# Towards Improving Faithfulness in Abstractive Summarization

**Xiuying Chen**[1]    **Mingzhe Li**[2]    **Xin Gao**[1,3*]    **Xiangliang Zhang**[4,1*]

[1] Computational Bioscience Reseach Center, King Abdullah University of Science and Technology
[2] Ant Group
[3] SDAIA-KAUST AI
[4] University of Notre Dame
{xiuying.chen, xin.gao}@kaust.edu.sa, limingzhe.lmz@antgroup.com,xzhang33@nd.edu

## Abstract

Despite the success achieved in neural abstractive summarization based on pre-trained language models, one unresolved issue is that the generated summaries are not always faithful to the input document. There are two possible causes of the unfaithfulness problem: (1) the summarization model fails to understand or capture the gist of the input text, and (2) the model over-relies on the language model to generate fluent but inadequate words. In this work, we propose a Faithfulness Enhanced Summarization model (FES), which is designed for addressing these two problems and improving faithfulness in abstractive summarization. For the first problem, we propose to use question-answering (QA) to examine whether the encoder fully grasps the input document and can answer the questions on the key information in the input. The QA attention on the proper input words can also be used to stipulate how the decoder should attend to the source. For the second problem, we introduce a max-margin loss defined on the difference between the language and the summarization model, aiming to prevent the overconfidence of the language model. Extensive experiments on two benchmark summarization datasets, CNN/DM and XSum, demonstrate that our model significantly outperforms strong baselines. The evaluation of factual consistency also shows that our model generates more faithful summaries than baselines[2].

## 1 Introduction

In recent years, text generation has made impressive progress [1, 2, 3]. The abstractive summarization task, aiming to produce a concise and fluent summary that is salient and faithful to the source document, has become a research hotspot due to its broad application prospect. The prevalence of pretrained transformer language models (LM) [4, 5] has largely improved the fluency and salience of generated summaries. However, studies [6, 7] showed that many summarization models suffer from unfaithfulness problem, *i.e.,* the generated summary is not entailed by the information presented in the source document. Durmus et al. [8] highlighted two notions of the unfaithfulness problem in summarization: one is the manipulation of information presented in the input document (*intrinsic errors*), and the other is the inclusion of information not inferable from the input (*extrinsic errors*).

The *Intrinsic error* problem is often caused by the failure of document level inference, which is necessary for abstractive summarization. Specifically, the summarization model has misinformation inferred from the input document because of an inadequate encoder that misunderstands the source

---

*corresponding author
[2]https://github.com/iriscxy/FES

semantic information and a poor decoder that cannot fetch relevant and consistent content from the encoder. Several recent summarization models were proposed from this perspective. For example, Wu et al. [9] proposed a unified semantic graph encoder to learn better semantic meanings and a graph-aware decoder to utilize the encoded information. Cao et al. [10] used contrastive learning to help the model be aware of the factual information. The second type of error, *extrinsic error*, is often introduced by excessive attention paid to the LM, which ensures fluency while neglecting to summarize the source document. For example, a LM is inclined to generate the commonly-used phrase "score the *winner*" while the correct phrase is "score the *second highest*" which is less frequently used. This type of error has been studied in the neural machine translation task [11], but has not been addressed in abstractive summarization.

To address these errors, we propose a novel Faithfulness Enhanced Summarization model (FES). To prevent the *intrinsic error* problem, we design FES in a multi-task learning paradigm, *i.e.,* completing encoding-decoding for the summarization task with an auxiliary QA-based faithfulness evaluation task. The QA task poses an additional reasoning requirement on the encoder to have a more comprehensive understanding on the key semantic meanings of the input document and learn better representations than working only for summarization. The QA attention on the key entities of the input can also be used to align the decoder state with the encoder outputs for generating a faithful summary. To address the *extrinsic error* problem, we propose a max-margin loss to prevent the LM from being overconfident. Concretely, we define an indicator of the overconfidence degree of the LM. The risk of outputting extrinsic error tokens with low prediction probabilities is mitigated by minimizing this overconfidence indicator.

We validate the effectiveness of our FES model by conducting extensive experiments on public benchmark CNN/DM [12] and XSum [13] datasets. Experimental results demonstrate that our faithfulness enhanced summarization model has superior performance on the ROUGE scores and improves the faithfulness of news summarization over several strong baselines.

Our main contributions can be summarized as follows. (1) We propose a faithfulness enhanced summarization model, which alleviates the unfaithfulness problem from the encoder side and decoder side. (2) Concretely, we propose a multi-task framework to enhance the summarization performance by automatic QA tasks. We also propose a max-margin loss to control the overconfident problem of the LM. (3) Experimental results demonstrate that our proposed approach brings substantial improvements over the most recent baselines on benchmark datasets, and can also improve the faithfulness of the generated summary.

## 2   Related Work

**Abstractive Summarization.** In recent years, the research on text generation has made impressive progress [14, 15], which promotes the progress of abstractive summarization. The abstractive summarization task generates novel words and phrases not featured in the source text to capture the salient ideas of the source text [16]. Most works apply an encoder-decoder architecture to implicitly learn the summarization procedure [17, 18]. More recently, applying pretrained language models as encoder [4, 19] or pre-training the generation process by leveraging a large-scale of unlabeled corpus [20, 21] brings significant improvements. Explicit structure modeling has also been shown to be effective in summarization tasks. For example, Jin et al. [22] incorporated semantic dependency graphs to help generate sentences with better semantic relevance, and Wu et al. [9] came up with a unified semantic graph to aggregate relevant disjoint context from the input.

**Fact Consistency for Abstractive Summarization.** Producing a summary that is entailed by the information presented in the source document is a key challenge in the summarization task, and less progress has been made on it. Pioneer works [23, 24] incorporated fact descriptions or entailment knowledge to enhance faithfulness. More recently, Zhu et al. [25] modeled the facts in the source article with knowledge graphs based on a graph neural network. Cao et al. [10] proposed to leverage reference summaries as positive training data and erroneous summaries as negative data, to train summarization systems that are better at distinguishing between them. Aralikatte et al. [26] introduced focus attention mechanism to encourage decoders to proactively generate tokens that are similar or topical to the input document. On the contrary, other works post-edit the generated summaries. Different from previous works, we enhance the semantic understanding of the document with faithfulness evaluation as a direct signal and prevent the overconfidence of LM.

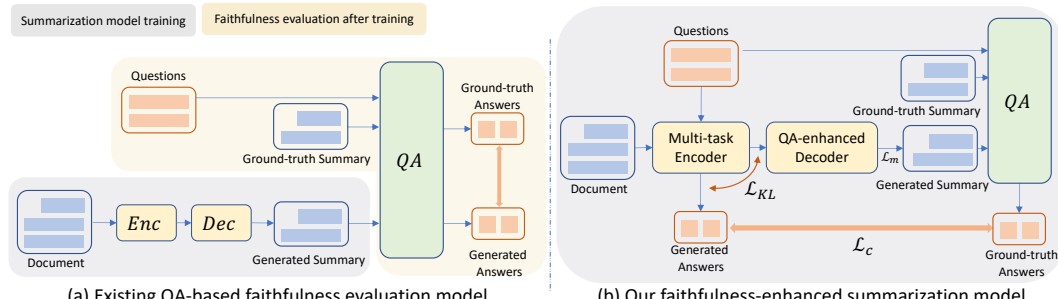

(a) Existing QA-based faithfulness evaluation model      (b) Our faithfulness-enhanced summarization model

Figure 1: The comparison of the existing QA-based faithfulness evaluation model and our faithfulness-enhanced summarization model. The QA task integrated in our model provides an auxiliary supervision signal to understand the document in the training process and enhance the faithfulness of the generated summary.

**Multi-task Learning.** Multi-task learning is a learning paradigm in machine learning and it aims to leverage useful information contained in multiple related tasks to help improve the generalization performance of all the tasks [27]. There is a large quantity of natural language processing tasks formulated by multi-task learning, such as word segmentation, POS tagging, dependency parsing, and text classification [28, 29, 30, 31]. In this work, we apply multi-task learning to summarization and question-answering tasks for faithfulness enhancement.

## 3 Methodology

### 3.1 Problem Formulation

For an input document $X = \{x_1, ..., x_{n_x}\}$, we assume there is a ground truth summary $Y = \{y_1, \ldots, y_{n_y}\}$. In our faithfulness enhanced setting, $n_q$ question answering pairs $Q = \{Q^1, ..., Q^{n_q}\}$ with corresponding answers $A = \{A^1, ..., A^{n_q}\}$ are also attached with $X$. In the training process, our model is given QA pairs and document-summary pairs. It tries to extract answers $A$ to the questions and generate the summary $Y$. In test stage, our model is given document $X$ and questions $Q$, and predicts the answers and summary. The final goal is to generate a summary that is not only informative but also consistent with document $X$.

Following, we introduce our proposed *Faithfulness Enhanced Summarization* model, which is generally built on Transformer [32]. The faithfulness enhancement is implemented from three aspects: (1) *Multi-task Encoder*. It improves the semantic understanding of the input document by examining the quality of the encoded document representations for an auxiliary QA task. The encoded representation thus captures the key inputs for making faithful summary. (2) *QA Attention-enhanced Decoder*. The attention from the multi-task encoder aligns the decoder with the encoder so that the decoder can fetch more accurate input information to generate the summary. (3) *Max-margin Loss*. This is a loss orthogonal to the generation loss. It measures the accuracy of the LM and prevents it from being overconfident in the generation process.

### 3.2 Multi-task Encoder

The multi-task encoder is designed for encoding the input document for both summarization and question-answering in an integrated training process, as shown in Figure 1(b). This is different from the previous work that uses QA in the post-generation stage for evaluating the faithfulness of the generated summaries [8, 7], as shown in Figure 1(a). We bring the QA closer to the encoder instead of leaving it for post-generated summary, and make the encoder be trained to accomplish the QA and summarization task in the meantime. This integrated training of a multi-task encoder includes faithfulness also as an optimization objective, besides the summary generation quality.

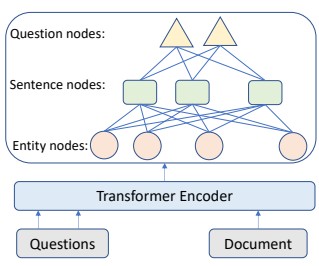

Figure 2: Multi-task encoder.

The answers are key entities from the document so that QA pairs focus on key information in the input.

As shown in Figure 2, we first apply the classic Transformer architecture to obtain token-level representations for the document and questions, denoted as $\mathbf{H}_w \in \mathbb{R}^{n_w \times d_e}$ and $\mathbf{H}_u \in \mathbb{R}^{n_q \times t_q \times d_e}$, where $n_w$ is the total number of tokens in the document, $n_q$ is the question number, $t_q$ is the token number in a question, and $d_e$ is the feature dimension. Then, we design the encoder to understand the question and the input document question from entity levels and sentence levels.

**Encoding at Multi-level Granularity.** We build the encoder by organizing the representation learning at different granularity levels. We use entities as the basic semantic unit as they contain compact and salient information across the document, and the reading comprehension questions focus on entities. Since a question is usually short, we create one node for each question. We add bidirectional edges from the questions to sentence nodes, and from sentence to entity nodes. These nodes act as the intermediary between sentences and enrich the cross-sentence relations. Because the initial directed edges are insufficient for learning backward information, we add reverse edges and self-loop edges to the graph following previous works [33]. We initialize node representations following the token level and word span level mean-pooling process [9].

Given the constructed graph with node features, we use graph attention networks [34] to update the representations of our semantic nodes. We refer to $\tilde{h}_i \in \mathbb{R}^{d_e}, i \in \{1, \cdots, (n_e + n_s + n_q)\}$ as the hidden states of input nodes, where $n_e$ and $n_s$ are the number of entity nodes and sentence nodes, respectively. The graph attention (GAT) layer is designed as follows:

$$z_{ij} = \text{LeakyReLU}\left(W_a\left[W_b\tilde{h}_i; W_c\tilde{h}_j\right]\right), \quad \alpha_{ij} = \frac{\exp(z_{ij})}{\sum_{l \in \mathcal{N}_i} \exp(z_{il})}, \quad l_i = \sigma(\sum_{j \in \mathcal{N}_i} \alpha_{ij}W_d\tilde{h}_j),$$

where $\mathcal{N}_i$ is the set of neighboring nodes of node $i$, $W_a$, $W_b$, $W_c$, $W_d$ are trainable weights and $\alpha_{ij}$ is the attention weight between $\tilde{h}_i$ and $\tilde{h}_j$. Besides, we add a residual connection to avoid gradient vanishing after several iterations: $h_i = \tilde{h}_i + l_i$. We iteratively use the above GAT layer and position-wise feed-forward layer [32] to update each node representation. The output entity feature matrix, sentence feature matrix, and question matrix, are denoted as $\mathbf{H}_e \in \mathbb{R}^{n_e \times d_e}$, $\mathbf{H}_s \in \mathbb{R}^{n_s \times d_e}$, and $\mathbf{H}_q \in \mathbb{R}^{n_q \times d_e}$, respectively.

**Answer Selector for the QA task.** After fusing information from the question and the document, we can select entities from the document as the answer to the question. Concretely, we apply the multi-head cross attention (MHAtt) between the question and the entities from the graph: $h_{qe}^i = \text{MHAtt}\left(h_e^i, \mathbf{H}_q, \mathbf{H}_q\right)$ to obtain question-aware entity representations, where $i$ is the question index. Based on the question-aware entity representations, we employ a feed-forward network (FFN) to generate the entity extracting probabilities $A^i = \text{FFN}(h_{qe}^i)$, where $A^i = (a_1^i, ..., a_{n_e}^i)$. The QA objective is to maximize the likelihood of all ground-truth entity labels $\hat{a}$:

$$\mathcal{L}_c = \sum_{i=1}^{n_q} \sum_{j=1}^{n_e} P\left(\hat{a}_j^i\right). \tag{1}$$

### 3.3 QA Attention-enhanced Decoder

A faithful decoder needs to attend to and fetch the important content from the encoder instead of mixing the inputs. We observe from §3.2 that the QA attentions on the key entities can be regarded as importance signals indicating which entities should be included in the summary. Hence, we propose a summary generator enhanced by QA attention. Generally, the decoder state attends to the encoder states with entities as intermediates, where the entity-level attention is guided by QA attentions.

Concretely, for each layer, at the $t$-th decoding step, we apply the self-attention on the masked summary embeddings $\mathbf{E}$, obtaining $u_t$. The masking mechanism ensures that the prediction of the position $t$ depends only on the outputs before $t$. Based on $u_t$, we then compute the cross-attention scores $c_t^e$ over entities.

$$u_t = \text{MHAtt}\left(e_t, \mathbf{E}_{<t}, \mathbf{E}_{<t}\right), c_t^e = \text{MHAtt}\left(u_t, \mathbf{H}_e, \mathbf{H}_e\right). \tag{2}$$

In effect, the first attention layer captures contextual features of the decoded sequence, while the second incorporates entity information in $c_t^e$. Herein, we minimize the bidirectional Kullback-Leibler

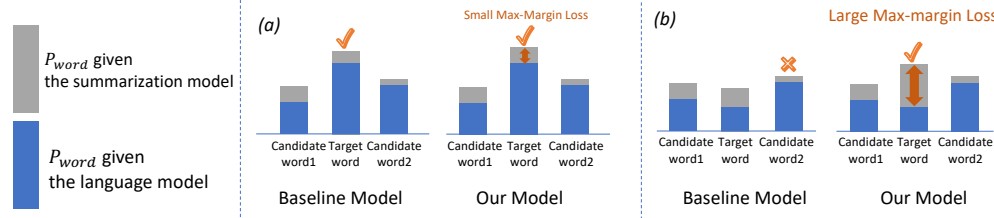

Figure 3: Max-margin loss for summarization model. (a) When the LM is accurate (the highest $P_{word}$ in blue for the target word), the correct target word is also predicted by both baseline and our summarization model with the highest $P_{word}$. (b) When the LM is not accurate enough (the highest $P_{word}$ in blue for a wrong candidate word2), our model can prevent the overconfidence of LM by max-margin loss and predict the correct target word, while the baseline model does not.

(KL) divergence between the QA attention $A^i$ and the summarization attention $E_t$ on the entities at the $t$-th step, to help the summarization model learn what entities are important:

$$\mathcal{L}_{KL} = \mathcal{D}_{KL}\left(\sum_{i=1}^{n_q} A^i \| \sum_{t=1}^{n_y} E_t\right), \tag{3}$$

where $n_y$ is the number of tokens in the ground truth summary. Note that this objective guides the cross-attention to capture correct contextual information rather than to only learn the QA attention distribution. So we only employ it on parts of attention heads to avoid "overfitting" to the QA task. We then use the entity-level attention to guide the selection of source tokens related to the key entities, by applying another MHAtt layer on source word sequence $\mathbf{H}_w$ and $c_t^e$:

$$v_t = \text{MHAtt}\left(c_t^e, \mathbf{H}_w, \mathbf{H}_w\right). \tag{4}$$

This context vector $v_t$, treated as salient contents summarized from various sources, is sent to an FNN to produce the distribution over the target vocabulary, *i.e.*, $P_t = \text{Softmax}\left(\text{FFN}(v_t)\right)$. All the learnable parameters are updated by optimizing the negative log likelihood objective function of predicting the target words:

$$\mathcal{L}_s = -\sum_{t=1}^{n_y} \log P_t\left(y_t\right). \tag{5}$$

### 3.4 Max-margin Loss

Previous works [35, 36] suggest that a poorly-informed decoder will neglect some source segments, function more as an open-ended LM, and thus will be prone to extrinsic errors. Inspired by faithfulness enhanced machine translation works [37, 38], we introduce a max-margin loss into the summarization task, for maximizing the difference of the predicted probability for each token of the summarization model and LM as shown in Figure 3, which suppresses the tendency of summarizer to generate common but unfaithful words. Concretely, we first define the margin between the summarizer and the LM as the difference of the predicted probabilities:

$$m_t = P_t\left(y_t \mid y_{<t}, X\right) - P_t^{LM}\left(y_t \mid y_{<t}\right), \tag{6}$$

where $X$ is the input document, and $P_t^{LM}$ denotes the predicted probability of the $t$-th token of the LM. Note that the language model has no access to the input document, and only takes the decoded summary prefix as input. Intuitively, if $m_t$ is large, then the summarization model is apparently better than the LM. When $m_t$ is small, there are two possibilities. One is that both the LM and the summarization model have good performance, hence the predicted probabilities should be similar. The other possibility is the LM is not good enough but overconfident, which leads to a summarizer with poor performance.

Hence, we present the max-margin loss $\mathcal{L}_m$, which adds a coefficient to the margin:

$$\mathcal{L}_m = \sum_{t=1}^{n_y}\left(1 - P_t\right)\left(1 - m_t^5\right)/2, \tag{7}$$

where we abbreviate $P_t(y_t \mid y_{<t}, X)$ as $P_t$. The term $(1 - m_t^5)/2$ is a non-linear monotonically decreasing function in regard to $m_t$, which ensures the optimization of maximizing $m_t$. We choose Quintic function (fifth power) here as it is shown to be more stable [38]. The first factor $(1 - P_t)$

is for fitting the two possibilities we discussed above. When $P_t$ is large, the summarization model learns the $y_t$ well and does not need to pay too much attention on $m_t$. This is reflected by $(1 - P_t)$, a small coefficient of $m_t$. On the other hand, when $P_t$ is small, it means that the summarizer needs to be better optimized, and a large coefficient $(1 - P_t)$ enables the model is able to learn from the margin information.

The above four losses, $\mathcal{L}_c$, $\mathcal{L}_s$, $\mathcal{L}_{KL}$, and $\mathcal{L}_m$ are orthogonal and can be combined to improve faithfulness.

## 4 Experiment

### 4.1 Experimental Setup

**Datasets.** We demonstrate the effectiveness of our approach on two public datasets, CNN/DM and XSum, which have been widely used in previous summarization works. Both datasets are based on news and consist of a large number of events, entities, and relationships that can be used to test the factual consistency of summarization models.

Note that our summarization model is accompanied by a QA task. Hence, we pre-construct QA pairs for each case using QuestEval tool provided by Scialom et al. [7]. Concretely, QuestEval first selects a set of the named entities and nouns as answers from the source document. Then, it uses a finetuned answer-conditional question generation T5 [39] model to generate questions via beam search. To ensure the quality of the QA pairs, we only select those questions for which the question-answering model [7] gives the right answers. Finally, we take 38 QA pairs for CNN/DM and 27 pairs for XSum on average. For training the summarization model, intuitively, we want the questions to focus on the key information in the input. Hence, we select the pairs where the answers have the highest ROUGE-L scores with the target summaries as oracle pairs. A BART-based extraction model [21] is then trained to predict important answers (QA pairs). Dou et al. [40] showed that it brings more benefits when using the oracle guidance in the training phase. Hence, for the training dataset, we use the oracle QA pairs, *i.e.,* the first 8 pairs with the highest ROUGE scores as input. For validation and test, we use the pairs selected by the extraction model.

**Baselines.** We first compare our model with recent factual-consistent summarization models: (1) FASum [25] is a model that extracts and integrates factual relations into the summary generation process via graph attention. (2) CLIFF [10] leverages reference summaries as positive data and erroneous summaries as negative data to train summarization systems. We also compare our proposed model with recent abstractive summarization models: (3) BART [21] is a state-of-the-art abstractive summarization model pretrained with a denoising autoencoding objective. (4) PEGASUS [20] is a pre-training large Transformer-based encoder-decoder models for summarization task. (4) GSum [40] is a summarization framework that can take external sentence guidance as input. (5) SimCLS [41] bridges the gap between the learning objective and evaluation metrics by a reference-free evaluation.

**Implementation Details.** We implement our experiments in Huggingface [42] on 4 NVIDIA A100 GPUs. We build our models based on BART (facebook/bart-large) for CNN/DM and PEGASUS (google/pegasus-xsum) for XSum following their hyperparameter settings, as they obtain better performance on each dataset, respectively. The QA number is set to 8 unless otherwise stated. To avoid the model from learning the position information of entities or questions, we sort the entities and questions in alphabetical order. We use Adam optimizer with $\epsilon$ as 1e-8 and $\beta$ as (0.9, 0.999). The learning rate is set to 3e-5. The warm-up is set to 500 steps for CNN/DM and 125 for XSum. The batch size is set to 8 with gradient accumulation steps of 4. The beam size is set to 6 for CNN/DM and 8 for XSum. For pretrained LM models, we finetune the vanilla BART-based or PEGASUS-based LM on the CNN/DM and XSum dataset, respectively.

### 4.2 Main Results

**Automatic Evaluation.** We evaluate models using standard full-length ROUGE F1 [43]. ROUGE-1 (RG-1), ROUGE-2 (RG-2), and ROUGE-L (RG-L) refer to the matches of unigram, bigrams, and the longest common subsequence, respectively. We then use BERTScore [44] to calculate a similarity score between the summaries based on their BERT embeddings. We also evaluate our approach with the latest factual consistency metrics, FactCC [6] and QuestEval [7]. FactCC is a weakly-supervised, model-based approach for verifying factual consistency. QuestEval considers not

Table 1: Comparisons with state-of-the-art models on CNN/DM and Xsum. Marked ROUGE results are from [40, 41]. Numbers in **bold** mean that the improvement to the best baseline is statistically significant (a two-tailed paired t-test with p-value <0.01).

| Dataset | Model | Traditional Metric | | | Advanced Metric | | |
| --- | --- | --- | --- | --- | --- | --- | --- |
| | | RG-1 | RG-2 | RG-L | BERTScore | FactCC | QE |
| CNN/DM | FASum | 42.75 | 20.07 | 39.83 | 88.35 | 49.86 | 30.30 |
| | CLIFF | 44.16 | 21.13 | 41.06 | 88.62 | 51.58 | 33.28 |
| | BART* | 44.66 | 21.53 | 41.35 | 88.36 | 51.11 | 31.93 |
| | PEGASUS* | 44.17 | 21.47 | 41.11 | 88.27 | 50.98 | 31.84 |
| | GSum* | 45.94 | 22.32 | 42.48 | 88.64 | 53.51 | 33.98 |
| | SimCLS* | 46.67 | 22.15 | 43.54 | 88.78 | 53.21 | 33.83 |
| | FES | **46.91** | **22.84** | 43.47 | **89.54** | **55.23** | **35.50** |
| | FES(oracle) | 50.50 | 26.41 | 46.97 | 90.01 | 59.25 | 39.11 |
| | FES w/o multi | 44.95 | 21.86 | 41.61 | 88.93 | 54.18 | 34.90 |
| | FES w/o QA attention | 46.53 | 22.49 | 43.12 | 89.27 | 54.85 | 35.22 |
| | FES w/o margin | 46.23 | 22.26 | 42.82 | 89.22 | 54.60 | 34.93 |
| | FES w/o key | 45.50 | 22.25 | 42.12 | 89.03 | 54.47 | 34.78 |
| XSum | FASum | 42.18 | 19.53 | 34.15 | 90.46 | 17.27 | 19.01 |
| | CLIFF | 44.47 | 21.39 | 36.41 | 91.48 | 20.05 | 22.91 |
| | BART* | 45.51 | 21.94 | 36.75 | 91.32 | 19.91 | 20.36 |
| | PEGASUS* | 47.21 | 24.56 | 39.25 | 91.25 | 20.69 | 22.63 |
| | GSum* | 45.40 | 21.89 | 36.67 | 90.45 | 20.61 | 22.48 |
| | SimCLS* | 47.61 | 24.57 | 39.44 | 91.28 | 20.80 | 22.97 |
| | FES | **47.77** | **24.95** | **39.66** | **92.05** | **22.34** | **25.83** |

only factual information in the generated summary, but also the most important information from its source text, and finally gives a weighted F1 score QE.

The results are shown in Table 1. Among factual-consistent baselines, FASum performs relatively poorly. One possible reason is that FASum is not trained on pretrained models. CLIFF achieves better BERTScore and faithfulness scores than strong baseline BART. It can be seen that our model outperforms GSum by 0.97 ROUGE-1, 0.90 BERTScore on CNN/DM dataset and 3.35 QE score on XSum dataset, indicating questions in the multi-task provides better signals than important sentences. Finally, our model outperforms the best baseline SimCLS significantly in most of the metrics, especially in terms of faithfulness metrics (FactCC and QE), which proves the effectiveness of our methods. We also show the performance of our model on the test dataset when using the oracle QA pairs to evaluate the upper bound of the benefits brought by the QA task. We can see that oracles improve the performance significantly, with the best-performing model achieving a ROUGE-1 score of 50.50. The results indicate that 1) the model performance has the potential to be further improved given better QA pairs; and 2) the model does benefit from the auxiliary QA task.

**Human Evaluation.** Since automatic evaluations are not perfect and can be misleading sometimes, we further conduct a pairwise human evaluation to see whether our generated summaries are faithful to the source document. Following Cao et al. [10], we randomly sample 100 cases from CNN/DM and XSum, and then hire two fluent English speakers to evaluate summary informativeness (Inform.) and factual consistency (Factual.). For each article, the annotators are shown summaries generated by the baseline BART or PEGASUS model and two other systems. They then rate each system summary against the baseline summary. Next, the annotators are asked to label text spans with intrinsic and extrinsic errors. The compared models are baselines that achieve high automatic scores, and are shown without system names.

Table 2: Human evaluation: percentages of summaries that are worse than, tied with, or better than BART on CNN/DM dataset.

| Model | Inform. | | | Factual. | | |
| --- | --- | --- | --- | --- | --- | --- |
| | Lose↓ | Tie | Win↑ | Lose↓ | Tie | Win↑ |
| SimCLS | 9% | 75% | 16% | 7% | 85% | 8% |
| FES | **6%** | 72% | **22%** | 7% | 81% | **12%** |

Table 3: Human evaluation: percentages of summaries that are worse than, tied with, or better than PEGASUS on XSum dataset.

| Model | Inform. | | | Factual. | | |
| --- | --- | --- | --- | --- | --- | --- |
| | Lose↓ | Tie | Win↑ | Lose↓ | Tie | Win↑ |
| SimCLS | 7% | 86% | 7% | 8% | 79% | 13% |
| FES | **6%** | 83% | **11%** | **6%** | 78% | **16%** |

Table 2 and Table 3 show the manual evaluation results. Firstly, we can find that there are larger differences in terms of Informativeness on the CNN/DM datasets and in Factual consistency on XSum datasets. This echos the attributes of the datasets, where the summaries in CNN/DM are longer and cover more detailed information, while summaries in XSum are shorter and thus require the summarization model to have advanced summarization ability. Secondly, our model is more frequently rated as being more informative and more factual than SimCLS summaries on both datasets. This is consistent with our automatic evaluation metrics. The kappa statistics are 0.53 and 0.59 for informativeness and factual consistency respectively, indicating the moderate agreement between annotators. The statistical significance between FES and PEGASUS is tested using a two-tailed paired t-test for significance for $\alpha = 0.05$.

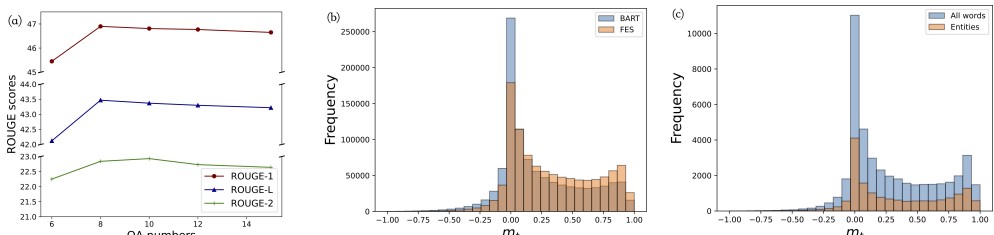

Figure 4: (a) ROUGE scores w.r.t. the number of used QA pairs on CNN/DM. (b) The distribution of margin $m_t$ of BART and FES. FES has $m_t$ much less negative, more around 0 and closer to 1, indicating the alleviation of overconfidence issue of LM. (c) The distributions of margin $m_t$ on entity words and all words. The proportion of entity words with $m_t$ around 0 is smaller, indicating that LM is accurate in predicting function words.

### 4.3 Discussions

**Ablation Study.** We perform an ablation study on CNN/DM dataset to investigate the influence of different modules in FES. First, we remove the multi-task framework to verify the effectiveness of joint learning on summarization. This means that the QA attention-enhanced mechanism in the decoder is also removed, so the model degrades to the BART summarizer with entity inputs and max-margin loss. Secondly, we keep the QA task but replace the QA attention-enhanced decoder with the vanilla Transformer decoder. Thirdly, we remove the max-margin loss to verify the influence of the overconfidence of the LM. Additionally, we randomly select QA pairs to see if the understanding of the model on the document has to be related to key information.

From the results in Table 1, we find without the multi-task framework, the performance of FES drops greatly by 1.86 RG-L score, 0.43 BERTScore, and 0.60 QE score on CNN/DM dataset, which indicates with the QA multi-task does strengthen the encoder to learn more comprehensive representations. Next, the QE score drops by 0.28 after the QA attention guidance is removed. This indicates that aligning the QA attention with the summarization attention on the important entities can help the model capture gist information from the input, and restrict such loss on a limited part of entities can guide the decoder to fetch meaningful content from the input. FactCC score drops by 0.63 after the max-margin loss is removed. It indicates that preventing the LM from being overconfident can help increase faithfulness. Finally, the performance of FES drops when using random QA pairs as guidance but outperforms BART by a large margin. This shows that enhancing the understanding of the document is helpful even when it is not always related to the key information. But the performance can be further improved by asking questions on key entities.

Table 4: Automatic evaluation results of the QA task.

| Model | EM | F1 |
|---|---|---|
| FES w/o multi | 76.34 | 83.59 |
| FES | **77.77** | **85.29** |

Table 5: The percent of $m_t < 0$ and average $m_t$ of our model and baseline.

| Model | Percent of $m_t < 0$ ($\downarrow$) | Average $m_t$ ($\uparrow$) |
|---|---|---|
| BART | 15.83% (ref) | 0.22 (ref) |
| FES | **13.50%** (-2.33%) | **0.33** (+0.11) |

**The Number of QA pairs.** To investigate how the number of QA pairs influences the performance of our model, we conduct experiments on the CNN/DM dataset with 6-14 oracle QA pairs. From results

Table 6: Examples of summaries generated by baseline models and our method. The **original fact**, intrinsic error, extrinsic error and the corresponding faithful fact in each summary are highlighted. *Italic* words are those predicted by the LM.

| Relevant QA pair | Relevant Context | Baseline Summary (truncated) | Our Summary (truncated) |
|---|---|---|---|
| Q: Who is the unexpected scorer of QPR's equaliser? A: Clint Hill | **Defender Clint Hill was the unexpected scorer of QPR's equaliser just after half-time to make it 2-2.** Hill grabs the Queens Park Rangers badge in celebration after scoring his first ever Premier League goal. | Christian Benteke equalised for the hosts just three minutes later. Clint Green put QPR ahead just after the half hour mark . | Defender Clint Hill scored his first ever Premier League goal to make it 2-2 just after half time at Villa Park. |
| Q: What did Ecuador issue after the Costa Rican government complained to the Ecuadorian authorities? A: An apology | The **Ecuadorain Ambassador Ricardo Patino then followed up with an apology** to Costa Rica and confirmed Ecuador had sent a letter to the government to settle the matter. | The stunt sparked outrage from Costa Rica, who complained to the authorities. Ecuador has since issued an apology and accepted the government's apology. | Costa Rican tourism minister said she was 'unhappy' with the use of her country's image. Ecuador has since issued an apology to Costa Rica and sent a letter of apology. |
| Q: Who scored one goal for Real Sociedad in the first half? A: Real Sociedad midfielder Gonzalo Castro | **Sociedad seized the lead once more in the 57th minute with a goal of the highest quality when Gonzalo Castro volleyed home Sergio Canales'cross. But Deportivo came roaring back once more** | Lucas Perez equalised for Deportivo La Coruna in the 40th minute. Gonzalo Castro scored the *winner* in the 57th minute for David Moyes ' side. | Gonzalo Castro scored one of the best goals of the season for David Moyes' side. Verdu Nicolas headed home to give Deportivo the lead with 12 minutes remaining. |
| Q: What season was he loaned back to Lille? A: 2014-15 season | **The Belgium striker was signed by Liverpool in a 10million deal after impressing at the 2014 World Cup in Brazil, before being loaned back to Lille for the 2014-15 season.** | Divock Origi joined Liverpool in a £ 10million deal from Lille this *summer*. The Belgium striker was loaned back to Lille for the 2014 - 15 season. | Origi was loaned back to Lille for the 2014-15 season after impressing at the World Cup. |
| Q: What two groups were at the height of tension? A: Asian and African americans | On Thursday, NPR headquartered in Washington, just 40 miles away from Baltimore ran its latest update on the urban turmoil that has erupted in the wake of the death of 25-year-old Freddie Gray | Ruben Navarrette: There 's little evidence that Asian businesses were targeted out of racial animus. | Ruben Navarrette: NPR report on Baltimore unrest focused on tension between African-Americans and Asians. |

in Figure 4(a), we see that the ROUGE score increases with the number of QA pairs, to begin with. After reaching 8 pairs, the improvements begin to vanish. One possible reason is that the answers no longer focus on the important information in the document. Note that the performance of FES remains at a high level in the range of 8-15 QA pairs, demonstrating the effectiveness and robustness of FES. At last, we choose to include 8 QA pairs in our model as default.

**QA Task Evaluation.** Since our framework also involves a question answering task, investigating the QA performance is helpful for understanding the model. Concretely, we use exact match (EM) and partial match (F1) to evaluate FES model and its ablation model, as shown in Table 4. Firstly, we can see that with the multi-task framework, both scores show significant improvements. This demonstrates that the two tasks can benefit each other, and the summarization task can also enhance QA performance. Secondly, the EM and F1 scores of QA are relatively high, showing that our model can also be used for answering questions on the document.

**Margin between FES and the LM.** We show the distribution of the margin $m_t$ defined in Eq. 6 from our FES and the BART in Figure 4(b). Firstly, there are still many tokens with negative $m_t$ and a large amount of $m_t$ around 0 for BART. This indicates that the LM is probably overconfident for many tokens, and addressing the overconfidence problem is meaningful for summarization. By comparison, it can be seen that the number of negative margin cases is significantly reduced in FES compared with BART. More precisely, we list the percentage of tokens with negative $m_t$ and the average $m_t$ for each model in Table 5. Compared with BART, FES reduces the negative $m_t$ by 2.33% and increases the average of $m_t$ by 0.11 points. This proves that the overconfidence problem of the LM is solved to a great extent. Besides, we draw the comparison of $m_t$ on all words and entity words in Figure 4(c). It can be seen that the proportion of around 0 for entity words is significantly reduced, which verifies our assumption that LM is accurate for many function words.

**Case Study.** We show several representative cases in Table 6, including the relevant QA pairs and relevant context from the source document, summaries generated by BART, PEGASUS, or SimCLS, and by our model on two datasets. The first three cases show summaries with intrinsic errors from baselines, because the baseline model misunderstands the source document. Our model makes faithful summaries with the help of answering the questions. For the forth case with extrinsic errors, the baseline summarization model and LM generate similar tokens, which are not included in the source document. FES has no extrinsic error, as it alleviates the overconfidence of the LM by introducing the max-margin loss. It is interesting to mention that the benefits brought by QA task and max-margin loss can complement each other. For example, in the third case, both the QA pair and max-margin loss prevent generating unfaithful information. We also show an error analysis in the last row. Both

generated summary include an unmentioned name, which can be found in the training dataset. A post-edit operation might solve the problem, and we look forward to improving it in the future.

## 5 Conclusion

In this paper, we propose the multi-task framework with max-margin loss to generate faithful summaries. The auxiliary question-answering task can enhance the model's ability to understand the source document, and the max-margin loss can prevent the overconfidence of the LM. Experimental results show that our proposed model is effective across different datasets. In the future, we aim to incorporate post-edit operation to improve faithfulness.

Generating faithful summaries is an important step toward real artificial intelligence. This work has the potential positive impact on an intelligent and engaging reading system. At the same time, if people rely too much on summarized systems of prompt reading, they may become less capable of reading long documents. Besides, the pre-training model may be injected with malicious and vulgar information, and results in server misleading summary. Therefore, we should be cautious of these advantages and disadvantages.

## Acknowledgments

We would like to thank the anonymous reviewers for their constructive comments. This work was supported by the SDAIA-KAUST Center of Excellence in Data Science and Artificial Intelligence (SDAIA-KAUST AI). This publication is based upon work supported by the King Abdullah University of Science and Technology (KAUST) Office of Research Administration (ORA) under Award No FCC/1/1976-44-01, FCC/1/1976-45-01, URF/1/4663-01-01, and BAS/1/1635-01-01. This work was also supported by Alibaba Group through Alibaba Research Intern Program.

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
