# OpenReview forum: "Towards Improving Faithfulness in Abstractive Summarization"
_NeurIPS.cc/2022/Conference — NeurIPS 2022 Accept_

### Official Review · Reviewer_Bgjw · 2022-07-10

**Rating:** 6
**Confidence:** 4
**Soundness:** 3 good
**Presentation:** 3 good
**Contribution:** 2 fair

**Summary:**

This paper proposes a multi-task model to enhance the faithfulness of abstractive summarization models. A synthetic QA dataset is created with the input documents as context and nouns/entities as answers. The model consists of a graph encoder that represents both the document and questions at various levels of granularity. The decoder is tuned by minimizing the KL divergence between the entity (QA) attention and the token (summary) attention. An additional max-margin loss is supplemented to the overall loss so that the language model does not overtake the decoding process. The model is evaluated on CNN/DM and XSUM and compared against a few baselines. Ablations are performed by iteratively removing the newly added components/losses.

**Questions:**

Questions/comments:
- It's usually the entity spans that are considered important. What about the relationships among them? For example, how will the model choose between "Argentina won against Mexico" versus "Mexico won against Argentina"? This problem may be further reinforced by the way it is implemented (Line 227: To avoid the model from learning the position information of entities or questions, we sort the entities and questions in alphabetical order.)
- An interesting experiment would be to plot the values of m_t only when the decoder generates entities. For most function words, it will likely be ~0 since the pre-trained decoder will be very confident while emitting them.
- Evaluating on additional metrics would make the results more convincing (entailment, FEQA, BertFaithful, etc.)
- Since the main objective of the multi-task setup is to get better entity representations into the decoder, wouldn't it be more natural to use NER as the auxiliary task? What benefits does QA bring over NER?

**Limitations:**

Other:
- Missing reference: Focus Attention: Promoting Faithfulness and Diversity in Summarization (https://research.google/pubs/pub50585/)

**Strengths And Weaknesses:**

In this line of research, it is difficult to exactly identify which change results in the largest performance jump and we see the same pattern here. Although it is common knowledge that attending to entities results in more faithful summaries, the paper presents a modeling-based solution to the same and does a good job of validating the solution. I would like to see the paper accepted, but have a few points of concern. I am willing to increase my rating provided all the questions/comments are addressed.

---

> ### Author Response · Authors · 2022-08-01
> **Response to Reviewer Bgjw**
>
> Thank you for the valuable comments that help us improve the work. Below we address the concerns mentioned in the review:
>
> *Q1: It's usually the entity spans that are considered important. What about the relationships among them? For example, how will the model choose between "Argentina won against Mexico" versus "Mexico won against Argentina"?*
>
> A1: Our model doesn’t directly learn the relationships between entities by encoding entity span, instead, it answers questions that focus on key entities. For the example you mention, our model might learn “Argentina won against Mexico” by answering the question “which team wins the competition?”.
>
> *Q2: An interesting experiment would be to plot the values of $m_t$ only when the decoder generates entities. For most function words, it will likely be ~0 since the pre-trained decoder will be very confident while emitting them.*
>
> A2: Thanks for your suggestion, and we draw the comparison of $m_t$ on all words and entity words which can be found in the Appendix.
> It can be seen that the proportion of $m_t$ around 0 for entity words is significantly reduced, which verifies our assumption that LM is accurate for many function words.
>
> *Q3: Evaluating on additional metrics would make the results more convincing (entailment, FEQA, BertFaithful, etc.)*
>
> A3: Following your suggestion, we test the FEQA scores of our model and baselines as shown below:
>
> | FEQA   | CNN/DM | XSum |
> |--------|--------|------|
> | BART   |   36.08     |   24.82   |
> | SimCLS |    36.95   |   24.87   |
> | FES    |    37.31   |  25.01    |
>
> We omit the entailment and BertFaithful evaluation since they are not publicly available tools and need extra training and finetuning.
>
> *Q4: Since the main objective of the multi-task setup is to get better entity representations into the decoder, wouldn't it be more natural to use NER as the auxiliary task? What benefits does QA bring over NER?*
>
> A4: We assume that the benefits of our multi QA task are not only to obtain better entity representations. What is more, we expect the summarization model to have a more comprehensive understanding of the document by answering questions. Hence, it is a document-level understanding of the key information, not just focusing on the attributes of words.
>
> Corresponding revision: We’ve added the introduction of missed citations in the revision and added the evaluation based on FEQA in the Appendix.
>
> We hope that these answers can address your questions. If you have future suggestions or questions about our paper, we will feel very happy to share more responses.

---

### Official Review · Reviewer_thb3 · 2022-07-11

**Rating:** 7
**Confidence:** 4
**Soundness:** 4 excellent
**Presentation:** 3 good
**Contribution:** 3 good

**Summary:**

Modifies state-of-the-art seq2seq summarization models (BART/PEGASUS) to be more faithful when generating summaries. There are a few orthogonal techniques/improvements: (1) adding QA auxiliary task to improve representations; (2) using a fancier encoder that creates representations for semantic nodes using Graph attention networks; (3) modifying decoder attention; (4) adding max margin loss comparing LM vs summarization token probabilities.

QA pairs are created for summarization data using QuestEval (Scialom et al); pairs with answers with high ROUGE overlap with summary work better.

Ablations of enhancements are done on the CNN/DM dataset, while final model is also evaluated on XSum. Automatic metrics involve ROUGE/BERTScore, FactCC, and QE. Human eval is done via pairwise comparison of final model with baseline PEGASUS/BART models, evaluating both informativeness and factuality. In the end the final model does best in both automatic and human eval evals.


**Questions:**

# Questions / Clarifications
- For the Max-margin loss, how do you train the LM? I don't think this is described.
- In line 209, you use oracle QA Pairs instead of those from the extraction model, which is used at test time. What is the effect on performance vs using the extraction model at training?
- Using QA as an auxiliary task seems intuitive for improving representations for summarization, but it is unclear in the paper how much of improvement is from using a fancier encoder (with graph attention). It'd be great to simply have a baseline with no architecture changes, but only adding the QA auxiliary task.
- Table 2 shows FES doing better than the baseline, but SimCLS also does better. Have you compared SimCLS and FES directly in a side-by-side?
- When training the semantic nodes, is the Transformer encoder frozen?
- In ablations, how can you remove 'multi' without removing QA attention. Doesn't the latter depend on the former since H_e is necessary for QA attention?


# Suggestions/comments
- Please add reference summaries to examples in Appendix
- It'd be nice to see the same ablations on Xsum that you did for CNN/DM. XSum has more faithfulness issues in general.
- It'd be great to compare the model and human references in a faithfulness side-by-side


**Limitations:**

While the ablations are in 4.3 are nice, they are only done with automatic metrics. it is unclear if all techniques are necessary to improve faithfulness in a human eval. e.g. multi-task encoder might be the only significant improvement. Which of the orthogonal techniques improves things by themselves (this is already shown in ROUGE on CNN/DM)?

**Strengths And Weaknesses:**

# Strengths
- Proposes multiple interesting techniques to improve faithfulness in summaries while also improving traditional evals e.g. ROUGE.
- Ablations done to show all contributed to improvements in ROUGE.
- Overall final model results are good and improves near-SOTA baseline models.
- Comprehensive evaluation was done using multiple automatic metrics and human eval.
- Faithfulness is a very important topic in summarization, and these improvements could have significant practical impact.

# Weaknesses
- Another task/dataset would be nice, e.g. non-news dataset, with known faithfulness issues, e.g. more abstractive. CNN/DM by its extractive nature doesn't typically exhibit huge faithfulness issues.
- Ablations would be better on XSum which has more faithfulness issues (more abstractive)
- Unclear whether all techniques are necessary to significantly improve faithfulness as judged by humans
- Some details of implementation appear to be missing (see Questions)

---

> ### Author Response · Authors · 2022-08-01
> **Response to Reviewer thb3**
>
> Thank you for the valuable comments that help us improve the work. Below we address the concerns mentioned in the review:
>
> *Q1: For the Max-margin loss, how do you train the LM? I don't think this is described.*
>
> A1: We finetune the vanilla BART-based or PEGASUS-based causal LM on the CNN/DM and XSum datasets, respectively. We’ve added this information in the revision.
>
> *Q2: In line 209, you use oracle QA Pairs instead of those from the extraction model, which is used at test time. What is the effect on performance vs using the extraction model at training?*
>
> A2: Using the extraction model at training will lead to significantly worse performance. Gsum[1] also shows a similar performance. This might be due to the weakened relevancy between QA pairs and references when using the extraction model at training. In such a case, the model will not learn to depend on the entity signals and thus be reduced to the original abstractive summarization baseline.
>
> *Q3: It'd be great to simply have a baseline with no architecture changes, but only adding the QA auxiliary task.*
>
> A3: The comparison of BART and BART with multi-task on CNN/DM dataset is shown below:
> |                 | RG-1  | RG-2  | RG-L  | BERTScore | FactCC | QE    |
> |-----------------|-------|-------|-------|-----------|--------|-------|
> | BART            | 44.66 | 21.53 | 41.35 | 88.36     | 51.11  | 31.93 |
> | BART+multi task | 46.01 | 22.18 | 42.65 | 89.17     | 54.50  | 34.84 |
>
> It can be seen that BART has a significant improvement on ROUGE-L and faithfulness score with multi-task.
>
> *Q4:Table 2 shows FES doing better than the baseline, but SimCLS also does better. Have you compared SimCLS and FES directly in a side-by-side?*
>
> A4: Following your suggestion, we directly compare SimCLS and FES on XSum dataset as shown below:
> |        |      | Inform |     |      | Factual |     |
> |--------|------|--------|:---:|------|-------|-----|
> | Model  | Lose | Tie    | Win | Lose | Tie     | Win |
> | FES vs SimCLS   |    6%  |   84%     |10%     |    5%  |        84% |    11% |
> It can be seen that our model still outperforms SimCLS.
>
> *Q6: When training the semantic nodes, is the Transformer encoder frozen?*
>
> A6: No, our model is an end-to-end model, and we train the model together.
>
> *Q7: In ablations, how can you remove 'multi' without removing QA attention. Doesn't the latter depend on the former since H_e is necessary for QA attention?*
>
> A7: As we said in line 281, when we remove multi-task, we also remove the QA attention, so the model degrades to the BART summarizer with entity inputs and max-margin loss.
>
> *Q8: Another task/dataset would be nice, e.g. non-news dataset, with known faithfulness issues, e.g. more abstractive.*
>
> A8: We choose these two datasets since they represent different dataset attributes, e.g., length, and abstractiveness, and they have been used in many faithfulness studies before [2,3,4]. Thanks for your suggestion, we are looking forward to trying our model on more datasets from different domains.
>
> *Q9: Ablations would be better on XSum which has more faithfulness issues (more abstractive).*
>
> A9: Following your suggestion, we provide the ablation study on XSum:
>
> |                      | RG-1  | RG-2  | RG-L  | BERTScore | FactCC | QE    |
> |----------------------|-------|-------|-------|-----------|--------|-------|
> | FES                  | 47.77 | 24.95 | 39.66 | 92.05     | 22.34  | 25.83 |
> | FES w/o multi        | 47.26 | 24.69 | 39.36 | 91.38     | 21.18  | 23.48 |
> | FES w/o QA attention | 47.69 | 24.81 | 39.57 | 91.84     | 22.06  | 25.39 |
> | FES w/o margin       | 47.49 | 24.74 | 39.40 | 91.60     | 21.55  | 24.49 |
>
>
> Corresponding revision: We’ve added the BART+multi task ablation results in Table 1 in the revision, the ablation study on XSum, and the direct comparison between FES and SimCLS in the Appendix.
>
> We hope that these answers have addressed your questions. If you have future suggestions or questions about our paper, we will feel very happy to share more responses.
>
> [1] GSum: A General Framework for Guided Neural Abstractive Summarization, NAACL 2021
>
> [2] CLIFF-Contrastive Learning for Improving Faithfulness and Factuality in Abstractive Summarization, EMNLP 2021
>
> [3] Enhancing factual consistency of abstractive summarization, AACL 2021
>
> [4] Factual error correction for abstractive summarization models, EMNLP 2020

---

> > ### Comment · Reviewer_thb3 · 2022-08-09
> > **thanks**
> >
> > Thanks for addressing my comments/questions.

---

### Official Review · Reviewer_GjUv · 2022-07-12

**Rating:** 7
**Confidence:** 4
**Soundness:** 4 excellent
**Presentation:** 4 excellent
**Contribution:** 3 good

**Summary:**

Generating summaries that are faithful is an important problem to be addressed for the summarization task. Faithfulness problem in summarization can be perceived in two types: (a) Intrinsic problem; and (b) Extrinsic problem. This paper tries to address these two types of problems with a novel modeling approach. This modeling approach proposed to use question-answering (QA) to examine whether the encoder efficiently represents the document thereby being able to answer the question on key information in the document. Further, the model is optimized on max-margin loss between language model and summarization model so that the model doesn’t over confidentially rely on the language modeling aspect and producing hallucinations.

Experimental results on CNNDM and XSum datasets suggest that the proposed model is indeed getting better on the factuality.


**Questions:**

1) Is the human evaluation gap statistically significant?
2) The proposed model is not 100% factual right? What are its limitations?


**Limitations:**

Adequately addressed the limitations.

**Strengths And Weaknesses:**

Strengths:
1) The proposed approach is very interesting and novel. The paper is well written and easy to follow.
2) Experimental results are thorough in terms of (a) comparing with many previous works and baselines, (b) showing the importance of each of the proposed modules in the model, and (c) showing the factuality improvements with human evaluations.
3) All the ablations are very interesting and very useful.

Weaknesses:
1) Human evaluations seem pretty close, with most of the weight going to Tie. Also, the sample size seems small given the close gap between the models. Can you increase the sample size and also do some statistical significance testing?
2) Suggestion: Please provide qualitative or quantitative analysis on where the proposed model makes factuality errors. Table 6 is great, but would be more useful to understand the limitations as well.

---

> ### Author Response · Authors · 2022-08-01
> **Response to Reviewer GjUv**
>
> Thank you for the valuable comments that help us improve the work. Below we address the concerns mentioned in the review:
>
> *Q1:Human evaluations seem pretty close, with most of the weight going to Tie. Is the human evaluation gap statistically significant?  Also, the sample size seems small given the close gap between the models. Can you increase the sample size and also do some statistical significance testing?*
>
> A1: Our human evaluation method follows Cao [1], and the sampling size is also larger than many other summarization evaluation experiments (20 in [2,3,4]).
> Following your suggestion, we expanded the human evaluation size to 150, and the evaluation results of comparison between FES and BART on XSum are shown below:
>
> |        |      | Inform |     |      | Factual |     |
> |--------|------|--------|:---:|------|---------|-----|
> | Model  | Lose | Tie    | Win | Lose | Tie     | Win |
> | SimCLS    | 6%     |  87%     |  6%   |    9%  |   77%      |  14%   |
> | FES    | 4%     |  85%     |  11%   |    5%  |   76%      |  19%   |
>
>
> To show the significance of these results, we also conduct the paired student t-test for $\alpha=0.05$. We obtain a p-value of 1.92e-02 and 1.06e-05 for informativeness and factual consistency, respectively, which demonstrates the significance of our results.
>
> *Q2:Please provide qualitative or quantitative analysis on where the proposed model makes factuality errors. What are its limitations?*
>
> A2: Thanks for your suggestion, and we add two error analyses as below:
> | Relevant Context                                                                                                                                                                                                                                                                                  | Baseline Summary (truncated)                                                                                                                                                                  | Our Summary (truncated)                                                                                                                                                                                          |
> |-------------|-----------|-------|
> | Fernandes, Rangers chairman, was watching the 3-3 draw at Villa Park from his iPhone away  from the game.                                                                                                                                                                                         | QPR drew 3-3 with Aston Villa at Villa Park on Saturday.  Christian Benteke scored a hat-trick to earn a point for Villa.                                                                     | QPR drew 3-3 with Aston Villa in their Premier League  clash on Saturday. QPR chairman Tony Fernandes was watching  the game from his iPhone.                                                                    |
> | Baltimore Unrest Reveals Tensions Between African-Americans And Asians |  Ruben Navarrette: There 's little evidence that Asian businesses were targeted out of racial animus. | Ruben Navarrette: NPR report on Baltimore unrest focused on tension between African-Americans and Asians. |
> |                                                                                                                                                                                                                                                                                                   |                                                                                                                                                                                               |                                                                                                                                                                                                                  |
>
> The first summary gives the wrong information that the event happens on Saturday, and the second one introduces the wrong speaker. The event time and the speaker are unknown in the document, but can be found in the training dataset. We are looking forward to improving it in the future. Thanks for your suggestion!
>
> Corresponding revision: We have updated the human evaluation results and added error analysis in the experiment section in the revised paper.
>
> We hope that these answers can address your questions. If you have future suggestions or questions about our paper, we will feel very happy to share more responses.
>
> [1]CLIFF-Contrastive Learning for Improving Faithfulness and Factuality in Abstractive Summarization, EMNLP 2021
>
> [2] Ranking Sentences for Extractive Summarization with Reinforcement Learning, NAACL 2018
>
> [3] Text Summarization with Pretrained Encoders, EMNLP 2019
>
> [4] Hierarchical Transformers for Multi-Document Summarization, ACL 2019

---

### Official Review · Reviewer_BYmM · 2022-07-12

**Rating:** 6
**Confidence:** 4
**Soundness:** 2 fair
**Presentation:** 3 good
**Contribution:** 2 fair

**Summary:**

The paper addresses the unfaithfulness problem in abstractive summarization with two proposed techniques: (1) using a question-answering model during training where a multi-task encoder encodes either the source document or questions generated from the document, and a QA-enhanced decoder generates either the summary or the generated answers, and (2) using a max-margin loss defined on the difference between the language model and the summarization model. Experiments show that the proposed model outperforms strong baselines on both summary informativeness and factuality evaluations.

**Questions:**

How do we interpret the FactCC and QE results? How high can we achieve here, given a perfect summary? If it is 100, are we still far off from producing abstractive and factual summaries, given XSum results?

**Limitations:**

None.

**Strengths And Weaknesses:**

Strengths
* The results are very good both in terms of informativeness (ROUGE and BertScore) and factuality (FactCC and QE).
* The use of a QA-enhanced model is intuitive given that questions in the faithful summary should be answerable given the source document as the context.

Weaknesses
* I find the methodology section to be very unclear. For one thing, the section could have been easier to understand if it included a problem setup formally defining the multi-task setup. It is not clear what are the differences in this setting from a normal summarization setting, where we are only given two sequences of tokens (one for the source and another for the summary). One example is that it is not clear if the corresponding answers to the given questions are also given during training, or if a QA/Q generation model is assumed to be available. Another thing that is unclear is how entity/question/sentence "nodes" are created given the tokens from the transformer encoder.
* I find the second contribution (the use of max-margin loss between the language model and the summarization model) to be counterintuitive in terms of improving the faithfulness of the summaries. Firstly, the summarization model is basically a fine-tuned version of the language model, how can we assume that the summarization model produces "adequate words" while the language model does not? If such "over-reliance" on the language model exists in LM-finetuned summarization models, shouldn't models not finetuned with LM produce more factual summaries? We know given previous work that this is not the case.

---

> ### Author Response · Authors · 2022-08-01
> **Response to Reviewer BYmM**
>
> Thank you for the valuable comments that help us improve the work. Below we address the concerns mentioned in the review:
>
> *Q1: The differences in this setting from a normal summarization setting.  it is not clear if the corresponding answers to the given questions are also given during training. How entity/question/sentence "nodes" are created given the tokens from the transformer encoder.*
>
> A1: Thank you for the question. In line 146, the model training is specified to include the corresponding answers to the given questions. To further clarify the problem setting and details of the model training process, we have included a problem formulation and a more detailed model introduction in the revision. The problem formulation section in the revision:
> > For an input document $X = \{x_1,  ..., x_{n_x}\}$, we assume there is a ground truth summary $Y = \{y_1,  \dots, y_{n_y}\}$. In our faithfulness enhanced setting, $n_q$ question answering pairs $Q = \{Q^1, ..., Q^{n_q}\}$ with corresponding answers $A = \{A^1,...,A^{n_q}\}$ are also attached with $X$. In the training process, our model is given QA pairs and document-summary pairs. It tries to extract answers $A$ to the questions and generate the summary $Y$. In the test stage, our model is given a document $X$ and questions $Q$, and predicts the answers and summary. The final goal is to generate a summary that is not only informative but also consistent with document $X$.
>
> Regarding the creation of entity/question/sentence "nodes", for entity nodes, we create them by using QuestEval to select entities and nouns from the document. For questions, we create one node for each question. For sentence nodes, we use nltk to obtain the sentence-level segmentation of the document, and create one node for each. For all kinds of nodes, we use the mean pooling of the corresponding token span to obtain the node representations. We’ve also detailed the introduction in the revision.
>
> *Q2: The use of max-margin loss between the language model and the summarization model is counterintuitive in terms of improving the faithfulness of the summaries. Firstly, how can we assume that the summarization model produces "adequate words" while the language model does not? If such "over-reliance" on the language model exists in LM-finetuned summarization models, shouldn't models not finetuned with LM produce more factual summaries?*
>
> A2: Thanks for your thoughtful concern about the max-margin loss. We would like to clarify here that the language model referred to in our paper is a standard language model (LM) that tries to predict the next token given the summary prefix ($P_t^{LM}\left(y_{t} \mid y_{<t}\right)$), but it has no access to the source document. It is fine-tuned by only accessing the summary, so that it can provide a good foundation for the summarization model. Hence, the fine-tuned LM may generate ungrounded and fluent sentences based on prefix tokens (the existence of “over-reliance”), but cannot generate accurate summaries based on the document content. On the other hand, the summarization model based on the fine-tuned LM generates the summary based on the source document information ($P_t\left(y_{t} \mid y_{<t}, X\right)$). Therefore, the LM-finetuned summarization model produces "adequate words", while the fine-tuned LM (accessing only the summary) has the "over-reliance" issue.
>
> *Q3: How do we interpret the FactCC and QE results? How high can we achieve here, given a perfect summary? If it is 100, are we still far off from producing abstractive and factual summaries?*
>
> A3: The score of FactCC is 0.57 and the QE score is 0.68 on ground truth XSum test dataset according to our experiments.
>
> FactCC is a classification model predicting whether the generated summary is faithful by taking the document and the summary as input. QE uses finetuned Question Generation (QG) to generate questions based on the document and then uses Question Answering (QA) model to answer the question based on the generated summary. The evaluation result of FactCC depends on the comprehensive understanding of the input document and summary. Similarly, the evaluation by QE depends on the QG&QA models that are hard to be perfect. Therefore, both manners are not perfect evaluations for faithfulness. Even when taking the ground truth summary as input, FactCC and QE cannot reach the full score of 1.00.
> However, the correlation of both metrics with human annotation is relatively high. This indicates that FactCC and QE can reflect the faithfulness of summaries, and a larger score means better quality. This is also why these two evaluation manners have been popularly used in faithfulness evaluation in tasks such as summarization, text simplification, and data2text.
>
> Thank you again for the insightful questions. We hope that our answers and revised paper have addressed your questions.  If you have future suggestions or questions about our paper, we will feel very happy to share more responses.

---

> > ### Author Response · Authors · 2022-08-07
> > **Response to Reviewer BYmM**
> >
> > Dear Reviewer,
> >
> > We appreciate a lot for your insightful review comments. Do you have any further comments on our paper?
> >
> > Regards.

---

> > > ### Comment · Reviewer_BYmM · 2022-08-08
> > > **Thanks for the response!**
> > >
> > > All the issues I have raised have been resolved. Will raise my score.

---

### Meta-Review · Area_Chair_Ymdi · 2022-08-26

**Recommendation:** Accept
**Confidence:** Certain

**Metareview:**

All reviewers agree that the method is interesting, the experiment results are thorough and significant. Multiple reviewers mentioned the paper would have been stronger by demonstrating performance on a dataset in another domain (i.e., not newswire), but still gave high scores. Given the consensus among reviewers, this paper should be accepted.

**Award:**

No

---

### Decision · Program_Chairs · 2022-09-14

Accept